# Seeing in the Dark: Benchmarking Egocentric 3D Vision with the Oxford Day-and-Night Dataset

**Zirui Wang**[*][†]    **Wenjing Bian**[*][†]    **Xinghui Li**[*][†]
**Yifu Tao**[‡]    **Jianeng Wang**[‡]    **Maurice Fallon**[‡]    **Victor Adrian Prisacariu**[†]

[*]Equal Contribution    [†]Active Vision Lab    [‡]Dynamic Robot Systems Group
University of Oxford

## Abstract

We introduce Oxford Day-and-Night, a large-scale, egocentric dataset for novel view synthesis (NVS) and visual relocalisation under challenging lighting conditions. Existing datasets often lack crucial combinations of features such as ground-truth 3D geometry, wide-ranging lighting variation, and full 6DoF motion. Oxford Day-and-Night addresses these gaps by leveraging Meta ARIA glasses to capture egocentric video and applying multi-session SLAM to estimate camera poses, reconstruct 3D point clouds, and align sequences captured under varying lighting conditions, including both day and night. The dataset spans over 30 km of recorded trajectories and covers an area of 40,000 m$^2$, offering a rich foundation for egocentric 3D vision research. It supports two core benchmarks, NVS and relocalisation, providing a unique platform for evaluating models in realistic and diverse environments. Project page: https://oxdan.active.vision/

## 1   Introduction

Intelligent wearable devices like smart glasses are gaining traction in the research community. Unlike bulky AR/VR headsets, their compact, lightweight design makes them more suitable for everyday use. To become as essential as smartphones, smart glasses must perform reliably across diverse environments, including challenging ones. A particularly tough scenario is outdoor low-light conditions, which uniquely degrade 3D vision tasks such as reconstruction, novel view synthesis (NVS), and visual localization due to poor signal-to-noise ratios. These tasks are key to interactive 3D experiences, yet current methods struggle in such settings. This highlights the need for a large-scale, egocentric 3D dataset tailored to low-light environments.

Existing 3D datasets, typically captured with handheld or vehicle-mounted cameras, provide diverse imagery but lack the combination of natural head motion, color, and full-day lighting variation, which are keys for all-day-long egocentric applications. Driving datasets like Oxford RoboCar [1] and CMU [2] offer large-scale, varied scenes including night, but are mostly limited to planar motion, unsuitable for agile head movements. Handheld datasets such as Cambridge Landmarks [3] and InLoc [4] offer more pose variation but limited lighting diversity. Aachen Day-Night [5] targets night-time localization but includes few night queries. LaMAR [6] provides egocentric day-night data, but its grayscale headset imagery limits suitability for color-dependent consumer applications.

To overcome limitations in existing datasets, we present Oxford-Day-and-Night, a large-scale video dataset captured across five locations in Oxford at various times of day. Spanning 30 kilometers and 40,000 m$^2$, it complements current datasets to provide a more comprehensive benchmark for 3D vision. This dataset is enabled by two key components: the Meta ARIA glasses and the Oxford Spires dataset [7].

Meta ARIA glasses are compact, sensor-rich devices equipped with grayscale and RGB cameras, IMUs, GPS, and more, enabling seamless and accurate data collection. Their built-in visual Simultaneous Localization and Mapping (SLAM) system ensures robust, multi-session camera tracking

39th Conference on Neural Information Processing Systems (NeurIPS 2025) Track on Datasets and Benchmarks.

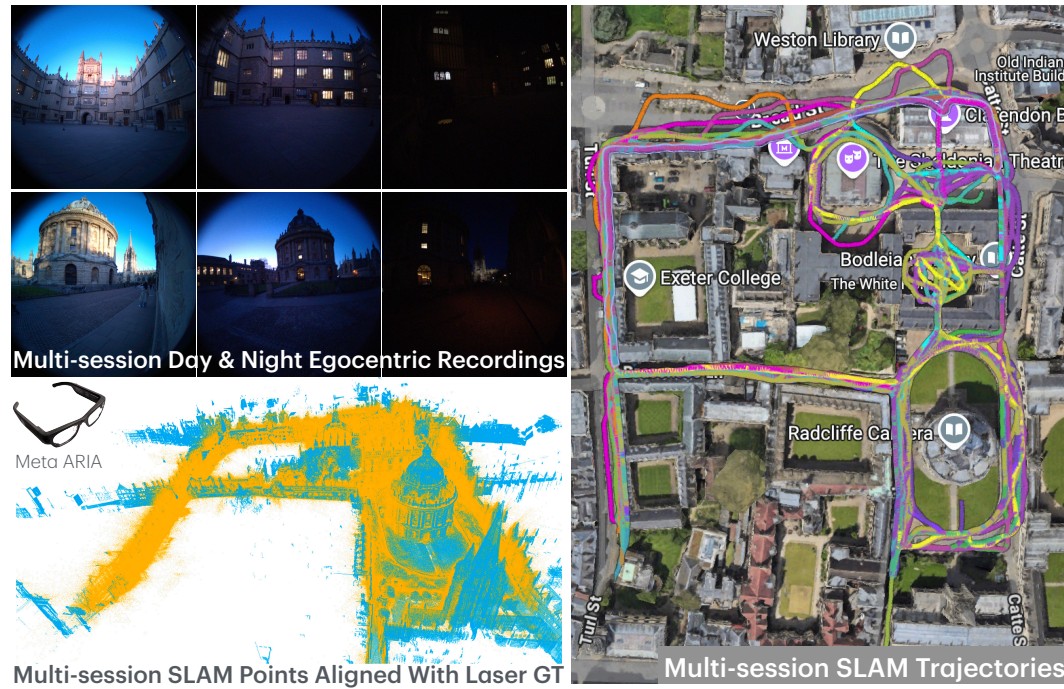

Figure 1: **Overview of the Oxford-Day-and-Night Dataset at Example Scene *Bodleian*.** Our dataset captures egocentric sequences across five locations in Oxford under diverse lighting conditions using Meta ARIA glasses. **Top-left**: Sample fisheye camera views across day and night recordings. **Bottom-left**: multi-session SLAM points aligned with high-quality laser ground truth. **Right**: Multi-session SLAM trajectories visualized on a satellite map, demonstrating consistent camera tracking across varying times of day. The dataset enables testing of challenging benchmarks for novel view synthesis and visual relocalization under extreme illumination changes.

and 3D reconstruction under dramatic lighting changes and city-scale settings. This multi-session SLAM system is the key component in creating our dataset, automating camera pose annotation for challenging night sequences at large scale. As a result, our video recordings cover 30 km and 40,000 m$^2$ areas in day and night settings, all paired with accurate camera poses and point cloud derived from the SLAM system.

Complementing this multi-session SLAM output, the Oxford Spires [7] dataset offers high-quality 3D laser scans of various Oxford locations. By aligning ARIA recordings with these scans, we both validate the accuracy of the ARIA data and offer reliable 3D geometric ground truth to support downstream tasks and benchmarking.

We benchmark two key 3D vision tasks using our dataset: novel view synthesis (NVS) and visual relocalization. For NVS, Oxford-Day-and-Night presents a challenging, city-scale setting with dramatic lighting variations, while the inclusion of ground-truth point clouds allows for quantitative evaluation of reconstructed geometry. For visual relocalization, the dataset offers a large set of nighttime query images (7197 in total), which is 37× larger than the Aachen night split (191 in total), enabling rigorous testing of localization pipelines under extreme conditions. Our experiments demonstrate that current state-of-the-art methods struggle on this dataset, exposing their limitations and underscoring the value of our benchmark.

Our contributions are summarized as follows. **First**, We present *Oxford-Day-and-Night*, a large-scale egocentric dataset featuring five urban scenes captured at multiple times of day with extreme illumination changes, along with their corresponding ground-truth point clouds. **Second**, We demonstrate two primary use cases: (i) a NVS benchmark for city-scale scenes with photometric diversity and geometry reference, and (ii) a visual relocalization benchmark featuring extensive night-time queries for testing robustness under challenging conditions. **Last**, We evaluate state-of-the-art NVS and relocalization methods on our dataset, revealing significant performance drops and underscoring the value of our dataset in future research.

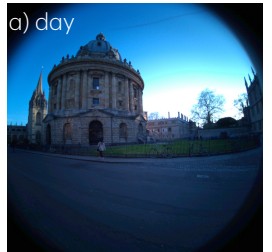 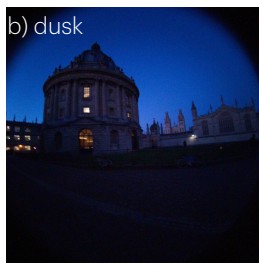 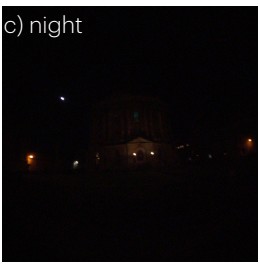 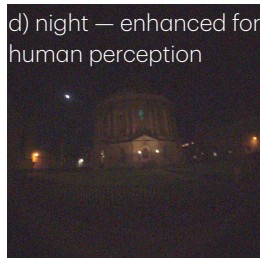

Figure 2: **Example Frames Captured at Different Lighting Conditions.** The severe degradation in visual quality from day to night highlights the difficulty of consistent scene understanding, posing significant challenges for both novel view synthesis (NVS) and visual relocalization methods.

## 2 Related Work

**3D Reconstruction Datasets.** Evaluating 3D reconstruction algorithms relies on accurate ground truth 3D models, which are typically obtained using methods such as SLAM, Structure-from-Motion (SfM)[8, 9], Terrestrial Laser Scanners (TLS)[10, 11], or through synthetic data [12]. Early multi-view stereo benchmarks like Middlebury [13] and DTU [11] used structured light scanners on robotic arms to capture small objects, while TLS has been employed for large-scale indoor and outdoor environments in datasets such as EuROC [10], ETH3D [14], Tanks and Temples [15], and ScanNet++[16]. Recent SLAM datasets[17, 18, 19] have extended TLS-based ground truth capture to outdoor settings, often integrating lidar for its robustness to lighting variation. These include environments ranging from natural landscapes [20] to structured urban areas [19, 21, 7].

Despite their geometric precision, many existing datasets depend on heavy, bulky, or sensitive equipment, which limits their ability to capture dynamic, agile camera motions, particularly from an egocentric perspective. Our dataset addresses this gap by integrating TLS-derived ground truth from Oxford Spires [7] with lightweight, wearable ARIA glasses. This combination enables high-fidelity 3D geometry alongside rich egocentric video sequences recorded under diverse motion patterns and lighting conditions, offering a valuable resource for advancing reconstruction under realistic and challenging scenarios.

**Novel View Synthesis Datasets.** NVS relies on datasets with multi-view images and accurate camera poses to enable the synthesis of novel viewpoints. Early datasets such as ShapeNet [22] and DTU [11] focused on object-centric settings, offering clean imagery and precise poses but limited diversity, often through synthetic renderings or controlled captures. As the field progressed toward more realistic scenarios, datasets like Tanks and Temples [15], ScanNet [9], and RealEstate10K [23] introduced real-world indoor and outdoor scenes with greater complexity in geometry and lighting. LLFF [24] and NeRF [25] established canonical benchmarks for neural rendering, with densely sampled forward-facing views, later extended to unbounded 360° captures in Mip-NeRF 360 [26].

More recent efforts have emphasized scale and diversity: CO3D [27] and Objaverse-XL [28] contribute large-scale object-centric data for real and synthetic domains, while scene-level datasets like Phototourism [29], MegaScenes [30], and DL3DV-10K [31] provide broader appearance variation across lighting, weather, and time. However, a consistent limitation remains, datasets with accurate ground-truth geometry are typically synthetic or limited in scale, while those offering visual diversity often lack high-quality geometry and precise camera calibration. Our dataset addresses this gap by combining large-scale real-world scenes, accurate ground-truth geometry, precise camera poses, and broad day-to-night visual variation, supporting the training and evaluation of generalizable NVS models under realistic conditions.

**Visual Relocalization Datasets.** Visual relocalization estimates a 6-DoF camera pose within a known environment using image data. Existing datasets for this task are typically categorized as indoor or outdoor, but each comes with notable limitations. Early indoor benchmarks such as 7-Scenes [32] and 12-Scenes [33] focus on small, static spaces with RGB-D input, but their constrained geometry and limited spatial coverage have led to performance saturation. Later efforts like InLoc [4], Indoor6 [34], and the Hyundai Department Store dataset [35] introduced more realistic conditions, featuring textureless surfaces, dynamic elements, and moderate illumination changes, but still fall

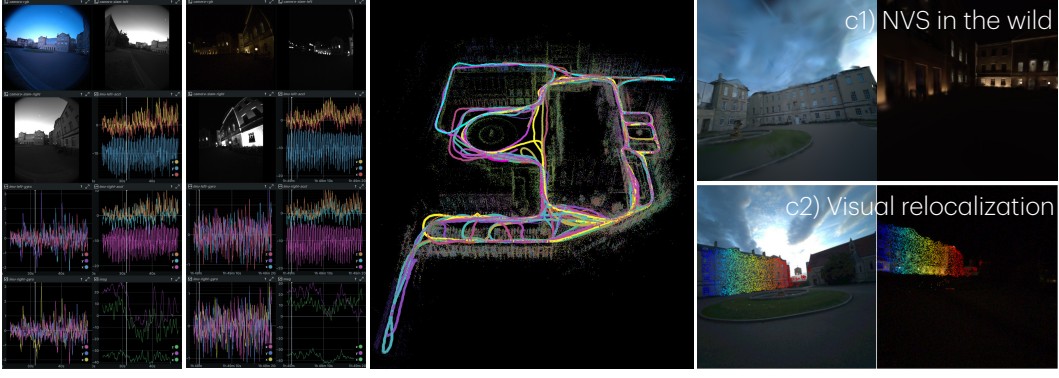

a) ARIA sensor stream ➡ b) MPS multi-session SLAM ➡ c) Benchmarks

Figure 3: **Data Collection and Processing Pipeline.** At a collection site, our pipeline starts with **a)** capturing 2–10 minute videos using ARIA glasses under varying lighting conditions. These multi-session recordings are processed using **b)** the MPS SLAM system to generate point clouds and camera trajectories in a unified coordinate frame. The colors of the points and trajectories represent different recording sessions; **c)** Leveraging the ARIA data and MPS outputs, we construct two dataset variants for NVS and visual relocalization tasks. Example scene: *Observatory Quarter*.

short in capturing the full variability needed for robust relocalization, particularly under extreme lighting shifts due to their reliance on artificial indoor lighting.

Outdoor datasets offer greater environmental diversity but often compromise in other areas. Vehicle-mounted datasets such as Oxford RoboCar [1], CMU [2], and KITTI [36] span large urban areas and varied conditions across time, weather, and lighting, yet are constrained to road-following, forward-facing viewpoints unsuitable for agile egocentric applications. Handheld datasets like Cambridge Landmarks [3] and InLoc provide more pose variety but limited lighting diversity. Aachen Day-Night [5] introduces night-time scenarios, though with relatively few queries. LaMAR [6] stands out for its egocentric, full-day data collection, but its grayscale headset imagery reduces relevance for color-dependent consumer applications. Overall, existing datasets lack the crucial combination of natural head motion, full-color imagery, and continuous day-long lighting variation required to rigorously evaluate robust, all-day, egocentric visual relocalization systems.

**Egocentric Datasets.** Popular egocentric datasets [37, 38, 39] have introduced collections of first-person videos in kitchen environments, annotated with fine-grained actions and object interactions. More recent efforts have expanded the scale, diversity, and realism of such data. Ego4D[40] represents a major milestone, offering large-scale, multimodal egocentric video with rich annotations for episodic memory, hand-object interaction, forecasting, and audio-visual understanding. EgoVid-5M[41] supports generative modelling with fine-grained action labels, kinematic data, and textual descriptions tailored for video generation tasks. Meta Project Aria has released several open datasets, including Aria Digital Twin[42], which provides high-fidelity ground truth for objects, environments, and human activities, and Aria Everyday Activities[43], which captures real-world tasks using RGB, stereo IR, IMU, eye-tracking, and audio sensors. EgoExo [44] stands out for offering synchronized egocentric and exocentric video recordings.

While existing datasets support action recognition, question answering, and general video under-standing, they often lack 3D geometry, camera motion, and lighting variation, particularly day–night transitions. In contrast, our large-scale dataset targets egocentric 3D vision under varying lighting conditions and includes camera poses and 3D point clouds aligned with ground truth geometry.

## 3 Oxford Day-and-Night

Our dataset is designed to advance research in egocentric perception under challenging, real-world conditions. It captures large-scale urban environments from a head-mounted, first-person perspective, characterized by natural and agile head movements. A key emphasis is placed on diverse lighting scenarios, with recordings conducted during the day, at dusk, and at night. For each site, the dataset includes high-quality video streams paired with estimated camera poses, along with a semi-dense

point cloud reconstructed via a SLAM system. From these fundamental elements, we derive two dataset variants tailored for NVS and visual relocalization tasks, each optimized for different image and point cloud density requirements.

## 3.1 Data Collection and Processing

We collect data using Meta ARIA glasses, which record raw sensor streams including IMU, RGB, and grayscale video. To capture varied lighting conditions, day, dusk, and night, sessions are recorded between 4-10pm, covering the natural transition from light to dark. Two individuals wear the glasses casually at each site. Recordings are grouped by location and processed with multi-session Machine Perception Service (MPS) provided by Meta, which estimates per-frame camera poses and semi-dense point clouds unified to a common coordinate frame. Fig. 3 illustrate this data collection process.

**Meta ARIA Glasses** is a lightweight, sensor-rich device designed for research-grade egocentric data capture. We use recording Profile 2, optimized for RGB video, capturing 20 FPS from both RGB (1408×1408, 110° FOV) and global shutter grayscale SLAM cameras (640×480, 150°×120° FOV), all with fisheye lenses for wide coverage. It also records high-frequency inertial data from dual IMUs (1000Hz and 800Hz). With 2 hours of runtime per charge, ARIA enables efficient, city-scale recording without bulky gear.

**MPS** is a cloud-based SLAM service that processes grayscale fisheye video and IMU data to generate high-frequency 6-DoF camera trajectories and semi-dense point clouds. It also support multi-session SLAM, which fuses recordings into a single global coordinate frame. This is the core component of our data collection pipeline, ensuring consistent spatial alignment across varying lighting conditions. The resulting globally aligned poses and 3D reconstructions form the backbone of our dataset.

## 3.2 NVS Dataset Creation

We preprocess video frames, camera poses, and semi-dense point clouds to support NVS tasks through three key steps. **First**, we temporal subsample video frames by $5\times$. As we recorded video at 20 fps, for large-scale scenes like *Bodleian* with 2.8 hours of footage, this results in more than 200,000 frames. While dense image input benefits NVS, such volume demands excessive storage and memory. **Second**, image undistortion, since ARIA uses fisheye lenses and most NVS methods assume a pinhole camera model, we provide both the original and undistorted images. **Third**, point cloud filtering, to improve geometric quality, we filter the semi-dense SLAM point cloud by removing points with high uncertainty, retaining only those with a depth standard deviation below $0.4$ m and inverse depth standard deviation below $0.005$ m$^{-1}$. This results in cleaner geometry suited for NVS systems such as 3DGS [45, 46, 47].

## 3.3 Visual Relocalization Dataset Creation

We construct our visual relocalization benchmark on top of our NVS dataset. Following established conventions [4, 5], the dataset comprises a set of daytime images with known camera poses (the database) and a separate set of images with unknown poses (the queries). The database images are used either to build a Structure-from-Motion (SfM) model for feature-matching-based relocalization methods [48, 49, 50], or as training data for pose regression-based approaches [51, 52]. Since each scene includes multiple video sequences recorded at different times, many frames depict the same locations from similar viewpoints, leading to redundancy. To promote diversity and reduce overlap, we apply spatial filtering based on the ground-truth camera poses.

We perform spatial filtering by first randomly shuffling all images in a scene and iterating through them to ensure pose diversity. An image is selected if its camera pose lies beyond a spatial radius of $\theta_{\text{pos}}$ from any previously selected pose; if nearby poses exist, the image is selected only if its orientation differs by at least $\theta_{\text{ori}}$. This guarantees diversity in both position and viewpoint. For outdoor scenes (Bodleian Library, H.B. Allen Centre, Keble College, Observatory Quarter), we use thresholds of $\theta_{\text{pos}} = 1.5$ meters and $\theta_{\text{ori}} = 20°$; for the indoor Robotics Institute scene, we adopt stricter thresholds of $\theta_{\text{pos}} = 0.5$ meters and $\theta_{\text{ori}} = 20°$ to reflect its smaller scale.

We apply spatial filtering independently to both daytime and nighttime images. From the filtered daytime set, we construct the visual relocalization benchmark by splitting the images into a database and a daytime query set using a 2:1 ratio. All filtered nighttime images are retained and used solely

Table 1: **Aria MPS Trajectory Quality.** We evaluate the aligned Aria trajectory quality using the point-to-point distance between the aligned MPS point cloud and the ground truth map.

| Point Dist. ↓ | Bodleian Library | H.B. Allen Centre | Keble College | Observatory Quarter | Robotics Institute |
|---|---|---|---|---|---|
| Mean (cm) | 9.7 | 5.2 | 9.3 | 7.1 | 2.4 |
| Median (cm) | 8.0 | 3.6 | 7.6 | 4.6 | 1.4 |

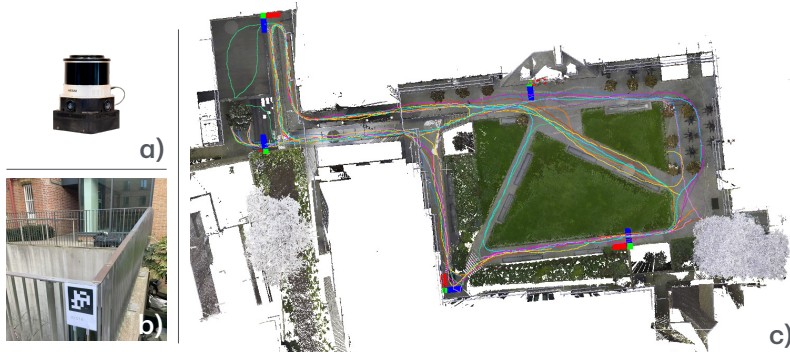

Figure 4: **ARIA MPS Quality Assessment.** We leverage *Frontier* and AprilTag to align ARIA recordings to TLS ground truth map. **a)** The *Frontier* handheld perception unit, equipped with three wide FoV cameras and a 64-channel LiDAR; **b)** A snapshot of an AprilTag; **c)** ARIA trajectories aligned within the ground truth TLS map in the *HBAC* scene. ARIA trajectories colors indicates from different recording sessions.AprilTag poses are highlighted with small colored coordinate frames.

as the nighttime query set, without further splitting. In total, the dataset comprises 5,466 database images, 2,819 daytime query images, and 7,179 nighttime query images. Full details of the filtering procedure are provided in the supplementary material.

### 3.4 Integration with Ground Truth Map from Oxford Sprires

Oxford Spires [7] is a high-fidelity dataset featuring precisely captured 3D point cloud maps using terrestrial laser scanning (TLS). We complement our Oxford day-and-night dataset with ground truth 3D point clouds from Oxford Spires to provide accurate point cloud reference models as the ground truth maps for benchmarking localization and NVS tasks. These maps were captured with a Leica RTC360 TLS, offering millimeter-level accuracy. We refer readers to [7] for more details.

### 3.5 ARIA MPS Accuracy Evaluation

To align the Aria world frame with a ground truth map, we developed an automated pipeline. AprilTags [53] are placed along planned paths, and their poses are logged in the ground truth frame using our handheld unit, *Frontier*, which captures images and LiDAR scans: images yield tag poses via AprilTag detection [54], while LiDAR scans are aligned with the map to produce centimeter-accurate trajectories [17]. Using calibrated camera-LiDAR extrinsics [55], all tag poses are expressed in the ground truth frame. We illustrate this process in Fig. 4.

Given the known tag poses in the ground truth map frame, each time a tag appears in the field of view of the Aria glasses, we compute the transformation between the ground truth map frame and the Aria world frame $\mathbf{T}_{\text{map,world}} = \mathbf{T}_{\text{map,tag}}(\mathbf{T}_{\text{aria,tag}})^{-1}(\mathbf{T}_{\text{world,aria}})^{-1}$, where $\mathbf{T}_{\text{map,tag}}$ is a tag pose in the ground truth map frame, and $\mathbf{T}_{\text{aria,tag}}$ is the individual tag detection in the local camera frame of the Aria glasses and $\mathbf{T}_{\text{world,aria}}$ is the corresponding Aria poses at the time of the tag detection in the arbitrary world frame from MPS. We discard detections with poor viewing angles or distances, then average valid transformations to align the closed-loop trajectory and point cloud with the map frame while preserving MPS output consistency.

To further improve MPS-to-GT alignment, the trajectory is refined by registering its associated point cloud to the ground truth map using Iterative Closest Point (ICP) [56]. The resulting trajectory is accurately aligned to the ground truth, with an average point-to-point error of $6.7\,\text{cm}$. Quantitative results are shown in Tab. 1.

## 3.6 Limitations

The accuracy of our ground-truth camera poses ultimately depends on the multi-session SLAM system provided by Aria MPS, and precisely quantifying SLAM accuracy in a large-scale environment is non-trivial. Traditional methods for obtaining ground-truth poses often rely on additional sensors, such as LiDAR or VICON motion capture systems. However, LiDAR can be unreliable in constrained areas like narrow tunnels, while VICON is impractical for city-scale deployments. Although we use AprilTags localized within our TLS maps for additional reference, both their detection and registration introduce further sources of error into the ground-truth estimation process.

# 4 Experiments

## 4.1 Benchmarking Visual Relocalization

**Benchmarked Methods.** We evaluate a broad range of visual relocalization methods on our dataset, including both *feature matching* (FM) approaches and *scene coordinate regression* (SCR) methods.

*Feature Matching Methods.* We adopt the HLoc pipeline [48], a widely used benchmark framework for structure-based localization. The pipeline begins by constructing a Structure-from-Motion (SfM) model using the daytime database images, based on pairwise image matching. At test time, the top 50 most visually similar database images are retrieved for each query image using NetVLAD [57], following standard practice. Feature matching is then performed between the query and retrieved images to establish 2D-3D correspondences via triangulated 3D points from the SfM model. Finally, the camera pose of the query image is estimated using PnP-RANSAC.

We evaluate four sparse matching methods within this pipeline: SIFT [58], SuperPoint [64] + SuperGlue [49] (SP+SG), SuperPoint + LightGlue [59] (SP+LG), and DISK [60] + LightGlue (DISK+LG). Additionally, we evaluate three recent dense matching methods: LoFTR [50], RoMA [61], and MASt3R [62], which directly compute dense correspondences between images without requiring keypoint detection.

*Scene Coordinate Regression Methods.* We also evaluate SCR-based methods, which directly regress 3D scene coordinates from 2D image pixels. Specifically, we test ACE [51], GLACE [52], and R-SCoRe [63]. These methods are trained on our daytime database images to predict per-pixel scene

Table 2: **Visual Relocalization Results on *Day* and *Night* Queries.** We report the percentage of query images correctly localized within three thresholds: (0.25m, 2°), (0.5m, 5°) and (1m, 10°). Results are shown for both feature-matching (FM) and scene coordinate regression (SCR) approaches. For FM approaches, the top 50 images retrieved using NetVLAD [57] are used for matching.

| | | **Visual Relocalization Results on *Day* Queries** | | | | |
|---|---|---|---|---|---|---|
| | | **Bodleian Library** | **H.B. Allen Centre** | **Keble College** | **Observatory Quarter** | **Robotics Institute** |
| FM | SIFT [58] | 91.91 / 96.34 / 97.02 | 75.95 / 81.65 / 82.91 | 84.98 / 88.78 / 91.06 | 89.86 / 92.69 / 92.92 | 70.07 / 73.07 / 74.56 |
| | SP+SG [49] | 96.26 / 98.85 / 99.16 | 96.84 / 98.73 / 99.37 | 94.68 / 97.34 / 98.10 | 94.81 / 95.99 / 95.99 | 89.28 / 90.77 / 91.77 |
| | SP+LG [59] | 95.73 / 98.32 / 98.85 | 96.84 / 98.10 / 98.10 | 92.78 / 96.20 / 97.15 | 94.34 / 95.75 / 95.99 | 88.28 / 89.53 / 90.02 |
| | DISK+LG [60] | 94.73 / 97.71 / 98.78 | 93.67 / 97.47 / 97.47 | 85.74 / 89.54 / 91.25 | 92.45 / 95.05 / 95.28 | 79.80 / 84.79 / 85.79 |
| | LoFTR [50] | 96.26 / 98.47 / 99.08 | 96.84 / 97.47 / 98.10 | 94.30 / 96.96 / 97.91 | 94.81 / 95.28 / 95.99 | 85.04 / 87.03 / 87.53 |
| | RoMA [61] | 92.14 / 95.42 / 96.34 | 87.34 / 93.04 / 94.30 | 91.83 / 96.20 / 97.15 | 91.27 / 93.87 / 93.87 | 85.79 / 87.78 / 88.53 |
| | MASt3R [62] | 90.61 / 93.82 / 96.18 | 94.30 / 98.73 / 99.37 | 94.68 / 97.91 / 98.86 | 89.39 / 92.92 / 94.58 | 84.54 / 90.52 / 94.02 |
| SCR | ACE [51] | 0.00 / 0.00 / 0.99 | 0.63 / 8.86 / 31.65 | 0.57 / 3.80 / 22.24 | 0.24 / 8.02 / 25.24 | 0.00 / 2.24 / 11.72 |
| | GLACE [52] | 0.00 / 0.61 / 10.38 | 0.63 / 4.43 / 34.81 | 0.19 / 4.18 / 35.93 | 0.24 / 6.13 / 33.02 | 0.00 / 0.75 / 29.43 |
| | R-SCoRe [63] | 47.71 / 68.32 / 79.62 | 50.00 / 64.56 / 73.42 | 60.46 / 75.10 / 85.74 | 45.52 / 58.02 / 71.23 | 5.99 / 12.47 / 18.20 |
| | | **Visual Relocalization Results on *Night* Queries** | | | | |
| | | **Bodleian Library** | **H.B. Allen Centre** | **Keble College** | **Observatory Quarter** | **Robotics Institute** |
| FM | SIFT [58] | 9.70 / 13.72 / 16.09 | 4.01 / 5.35 / 7.57 | 0.40 / 0.79 / 1.39 | 2.38 / 3.35 / 4.55 | 41.54 / 46.81 / 49.04 |
| | SP+SG [49] | 21.63 / 26.55 / 30.78 | 44.32 / 57.46 / 64.14 | 10.66 / 13.57 / 17.27 | 48.14 / 54.40 / 58.05 | 71.12 / 73.56 / 74.47 |
| | SP+LG [59] | 20.46 / 25.28 / 28.78 | 43.43 / 54.12 / 61.47 | 9.99 / 14.16 / 18.27 | 47.91 / 53.50 / 57.68 | 70.11 / 71.83 / 73.05 |
| | DISK+LG [60] | 14.75 / 17.54 / 20.12 | 9.58 / 11.36 / 14.70 | 0.53 / 0.79 / 1.06 | 16.77 / 20.42 / 22.95 | 53.19 / 57.55 / 60.49 |
| | LoFTR [50] | 22.42 / 26.55 / 29.20 | 41.20 / 52.78 / 58.80 | 10.39 / 13.63 / 17.01 | 50.00 / 57.00 / 60.13 | 66.87 / 70.52 / 72.44 |
| | RoMA [61] | 25.24 / 30.98 / 35.18 | 57.91 / 74.39 / 79.06 | 14.96 / 22.63 / 30.91 | 58.94 / 66.92 / 70.79 | 72.34 / 75.38 / 76.60 |
| | MASt3R [62] | 15.65 / 17.54 / 19.98 | 49.22 / 59.02 / 66.59 | 12.24 / 16.08 / 19.66 | 48.66 / 54.47 / 59.91 | 65.65 / 72.95 / 77.00 |
| SCR | ACE [51] | 0.00 / 0.00 / 0.00 | 0.00 / 0.00 / 0.00 | 0.00 / 0.00 / 0.00 | 0.00 / 0.00 / 0.00 | 0.10 / 0.10 / 0.91 |
| | GLACE [52] | 0.00 / 0.00 / 0.03 | 0.00 / 0.00 / 0.00 | 0.00 / 0.00 / 0.99 | 0.00 / 0.00 / 0.00 | 0.00 / 0.00 / 8.21 |
| | R-SCoRe [63] | 2.72 / 7.57 / 13.10 | 5.57 / 11.58 / 23.61 | 0.20 / 0.99 / 1.92 | 3.06 / 7.75 / 13.34 | 2.13 / 6.08 / 9.52 |

Table 3: **Accuracy of RoMA on the Visual Relocalization Dataset Using the HLoc Pipeline with Various Image Retrieval Methods on *Night* Queries.** We report the percentage of correctly localized query images within thresholds of (0.25m, 2°), (0.5m, 5°), and (1m, 10°).

| | Bodleian Library | H.B. Allen Centre | Keble College | Observatory Quarter | Robotics Institute |
|---|---|---|---|---|---|
| RoMA + NetVLAD 50 [57] | 25.24 / 30.98 / 35.18 | 57.91 / 74.39 / 79.06 | 14.96 / 22.63 / 30.91 | 58.94 / 66.92 / 70.79 | 72.34 / 75.38 / 76.60 |
| RoMA + DIR 50 [65, 66] | 33.46 / 39.10 / 42.30 | 55.46 / 72.16 / 81.51 | 16.48 / 23.36 / 28.33 | 56.33 / 65.28 / 69.90 | 74.27 / 78.52 / 79.84 |
| RoMA + OpenIBL 50 [67] | 43.95 / 51.24 / 54.50 | 60.36 / 73.50 / 78.62 | 18.66 / 27.47 / 35.67 | 59.09 / 66.32 / 70.19 | 71.73 / 75.18 / 76.19 |
| RoMA + MegaLoc 50 [68] | 70.25 / 79.09 / 82.22 | 66.37 / 81.51 / 87.31 | 31.50 / 42.22 / 51.82 | 72.06 / 80.92 / 84.50 | 78.22 / 82.27 / 83.69 |
| RoMA + GT Pose 20 | 80.57 / 89.58 / 92.88 | 68.82 / 81.51 / 85.75 | 41.50 / 57.78 / 71.01 | 80.55 / 87.48 / 90.98 | 84.50 / 89.36 / 90.48 |

coordinates. At inference time, they provide dense 2D-3D correspondences for each query image, from which the camera pose is estimated using PnP-RANSAC.

**Results.** We summarize the results of the evaluation on daytime and nighttime queries in Tab. 2, where we report the percentage of query frames with pose errors of within three thresholds: (0.25m, 2°), (0.5m, 5°) and (1m, 10°). Our experiments are conducted using 48GB NVIDIA RTX A6000 GPUs, with mapping times ranging from a few minutes to several hours, depending on the relocalization method and scene complexity.

**Analysis.** We observe that feature-matching (FM) methods significantly outperform scene coordinate regression (SCR) approaches on both daytime and nighttime queries, with some SCR methods failing entirely at night. This is consistent with the Aachen Day-Night benchmark [5], where SCR methods generally struggle in large-scale environments and under severe illumination changes. The performance gap between day and night is even more pronounced in our dataset, due to increased lighting variability that makes regressing consistent 3D coordinates especially difficult. FM methods perform well on daytime queries, likely because the query and database trajectories are similar, reducing viewpoint variation, but their performance drops notably at night, highlighting the challenge of low-light conditions. Among them, RoMA achieves the highest overall accuracy and is thus used for a deeper analysis of image retrieval, a factor often overlooked in favor of default choices like NetVLAD in the HLoc pipeline.

To evaluate retrieval quality, we pair RoMA with four retrieval methods: NetVLAD, DIR [65, 66], OpenIBL [67], and MegaLoc [68], retrieving the top 50 database images. A quasi-upper bound is also included using the 20 nearest images based on ground-truth poses. As shown in Tab. 3, more advanced retrieval methods significantly boost performance, with RoMA exceeding 80% accuracy under the strictest threshold (0.25m, 2°) when paired with ground-truth retrieval, indicating that retrieval, not matching, is the main accuracy bottleneck. However, RoMA's major limitation is its runtime: approximately 1 second per image pair, about 30× slower than SuperPoint+LightGlue (~0.03s). These findings point to two important research directions enabled by our dataset: (1) enhancing image retrieval under challenging conditions and (2) accelerating high-accuracy matchers like RoMA, where speed is the limiting factor.

## 4.2 Benchmarking NVS

**Benchmarked Methods.** We evaluate two state-of-the-art in-the-wild neural view synthesis (NVS) methods: Splatfacto-W [46] and Gaussian-Wild [47]. To train these models on our NVS dataset, we apply two preprocessing steps. First, we further subsample the image collections for each scene to approximately 2,500 images to ensure manageable CPU memory usage. Second, we perform voxel downsampling of the semi-dense point cloud using a voxel size of $0.1m$ ($0.2m$ for the *Bodleian* scene). Since GPU memory consumption is proportional to the number of initial 3D points, this step helps keep GPU usage under 80GB for these NVS systems.

**Result.** We follow the standard convention of selecting every 8th image as a test image and report image quality using PSNR and LPIPS metrics. For geometry evaluation, we utilize the ground truth 3D point clouds from Oxford-Spires [7] and measure the point-to-point distance between the centers of the 3DGS Gaussian primitives and the ground truth 3D maps. We compute the point to point distance using CloudCompare. The results are presented in Tab. 4 and Fig. 5.

In this experiment, Splatfacto-W outperforms Gaussian-Wild on the *H.B. Allen Centre* scene but underperforms on the remaining four scenes. However, as indicated by the LPIPS scores, both methods exhibit limited performance across these four scenes. This is primarily due to the large-scale

Table 4: **3DGS In-the-Wild Results.** We report image rendering and geometry quality using the following metrics: PSNR (↑) / LPIPS (↓) / point-to-point distance (↓). The 3DGS geometry is derived by extracting the centers of all Gaussian primitives, with point-to-point distance (meter) computed against the ground truth laser-scanned point cloud. Symbol "-" denotes the system produces a degenerated point cloud (less than 2000 gaussians after training).

| Method | Bodleian Library | H.B. Allen Centre | Keble College | Observatory Quarter | Robotics Institute |
|---|---|---|---|---|---|
| Splatfacto-W [46] | 25.98 / 0.60 / - | 25.65 / 0.59 / 0.75 | 27.96 / 0.59 / - | 25.83 / 0.63 / 0.36 | 22.73 / 0.61 / 0.42 |
| Gaussian-Wild [47] | 28.38 / 0.56 / 1.44 | 24.94 / 0.59 / 1.48 | 30.92 / 0.56 / 0.69 | 28.57 / 0.60 / 0.69 | 25.05 / 0.57 / 0.76 |

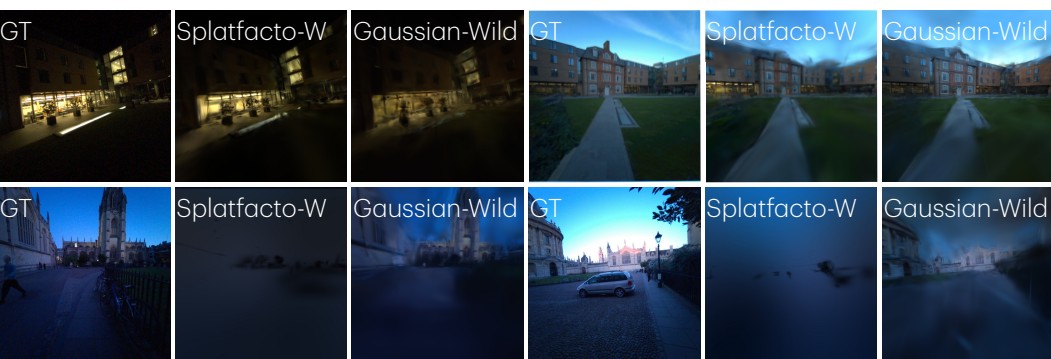

Figure 5: **NVS In-the-Wild Results in the H.B. Allen Centre (*top*) and Bodleian Library (*bottom*).** Compared to Gaussian-Wild, Splatfacto-W performs better at the H.B. Allen Centre but fails at the Bodleian Library. Although Gaussian-Wild produces some renderings with recognizable content, the overall quality is limited. These results highlight that current state-of-the-art NVS-in-the-wild methods still face significant challenges in large-scale environments with dramatic lighting variations.

nature of the dataset and the extreme lighting variations, ranging from daylight to poorly illuminated night conditions.

**Discussion.** *First*, further downsampling: as described in Sec. 3.2, we initially downsampled videos and filtered noisy 3D points to create an NVS dataset suitable for *future* use. However, *current* state-of-the-art in-the-wild 3DGS systems still struggle with this data scale. Therefore, we apply more aggressive temporal subsampling and spatial downsampling of the point clouds to ensure feasibility. *Second*, PSNR fails to reflect image quality. In Tab. 4, both methods achieve PSNR $> 25$ across most scenes, yet high LPIPS values $> 0.5$ reveal poor visual quality. Similar issues are observed with SSIM, as shown in the supplementary material. *Third*, point-to-point distance may offer a rough indication of NVS performance, but only when basic shapes are preserved and points are not aggressively culled during 3DGS optimization. More details can be found in supplementary.

## 5  Conclusion

Oxford Day-and-Night fills a crucial gap in egocentric 3D vision research by providing a large-scale, lighting-diverse dataset explicitly designed for challenging outdoor conditions, including nighttime scenarios. Through its combination of rich sensor data, robust multi-session SLAM annotations, and alignment with high-fidelity ground-truth geometry, the dataset enables rigorous benchmarking of novel view synthesis and visual relocalization methods at city scale. Our experiments reveal substantial performance degradation of current state-of-the-art approaches, particularly under extreme lighting changes, underscoring both the difficulty of the tasks and the value of our benchmarks. By exposing these limitations, Oxford Day-and-Night offers a powerful platform to drive progress in robust, all-day egocentric perception systems.

**Acknowledgement.** This research is supported by multiple funding sources, including an ARIA research gift grant from Meta Reality Lab, a Royal Society University Research Fellowship (Fallon), the EPSRC C2C Grant EP/Z531212/1 (TRO), and a National Research Foundation of Korea (NRF) grant funded by the Korean government (MSIT) under grant number RS 2024 00461409.

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

# Seeing in the Dark: Benchmarking Egocentric 3D Vision with the Oxford Day-and-Night Dataset (Supplementary)

https://oxdan.active.vision/

## A    Full Dataset Statistics

We collected our dataset across five locations in Oxford by walking while wearing ARIA glasses. The data collection took place over the course of one month. During this period, two collectors wore ARIA glasses and walked randomly within each collection site. In total, the walking trajectory spans 30 kilometers, includes 7 hours of walking, and covers an area of 40,000 $m^2$.

Detailed dataset statistics are provided in Tab. 5. Notably, our dataset offers a well-balanced distribution of day and night recordings, with an approximately 1:1 ratio. The covered areas and walking trajectories are visualized in Figs. 6 and 7.

Table 5: **Dataset Statistics.** We present a summary of the number of frames in the recorded videos, the NVS data variant (obtained by subsampling the video by $5\times$), and the visual relocalization data variant (with additional spatial subsampling and splitting into database, daytime queries, and nighttime queries). We also report the recording durations, trajectory lengths, and mapped area sizes.

| Scene | # Video Fr | # NVS Img | # Visual Reloc Img | | | Duration (hh:mm) | | | Trajectory Len (m) | | | Area ($m^2$) |
|---|---|---|---|---|---|---|---|---|---|---|---|---|
| | D & N | D & N | DB | Day Q | Night Q | Day | Night | D & N | Day | Night | D & N | D & N |
| Bodleian Lib. | 205405 | 41081 | 2542 | 1310 | 2908 | 01:32 | 01:18 | 02:50 | 7170 | 5617 | 12787 | 25939 |
| H.B. Allen Cen. | 29340 | 5868 | 305 | 158 | 449 | 00:13 | 00:10 | 00:24 | 975 | 765 | 1740 | 1271 |
| Keble College | 112205 | 22441 | 1020 | 526 | 1511 | 00:46 | 00:46 | 01:33 | 3574 | 3400 | 6974 | 5709 |
| Obs. Quarter | 87210 | 17442 | 821 | 424 | 1342 | 00:34 | 00:38 | 01:12 | 2853 | 3050 | 5903 | 5950 |
| Robotics Inst. | 57590 | 11518 | 778 | 401 | 987 | 00:25 | 00:22 | 00:47 | 1249 | 1030 | 2279 | 600 |
| Total | 491750 | 98350 | 5466 | 2819 | 7197 | 03:32 | 03:16 | 06:48 | 15822 | 13862 | 29685 | 39469 |

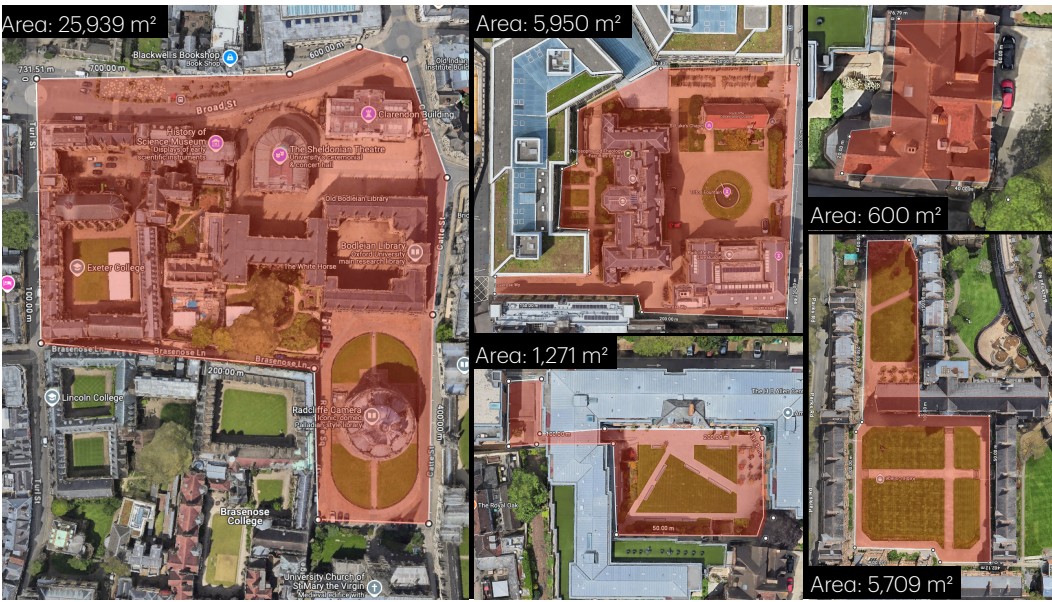

Figure 6: Our dataset covers 40,000 $m^2$ area.

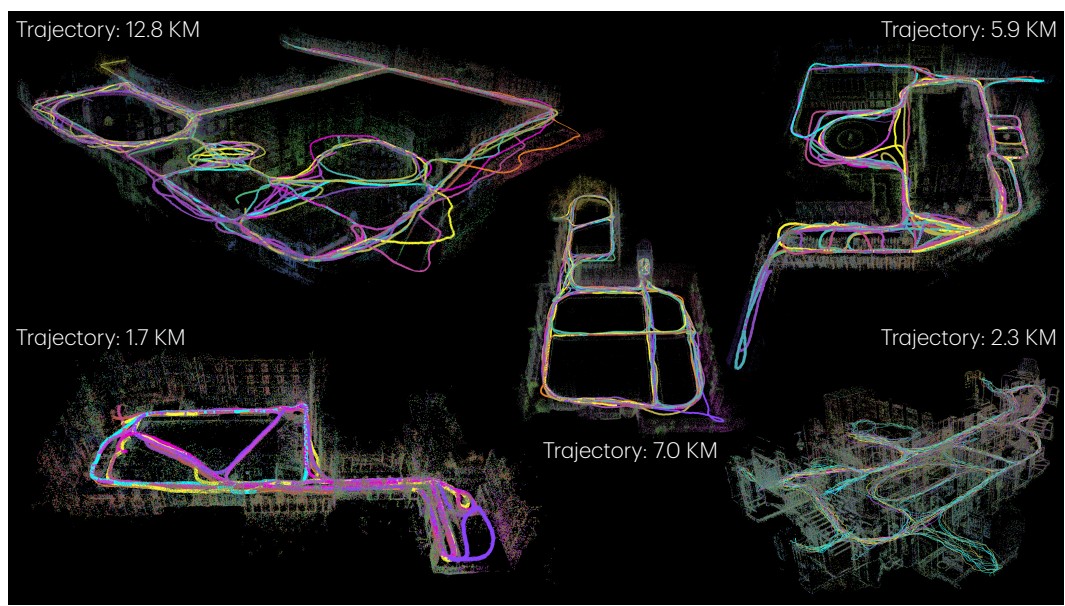

Figure 7: Our dataset spans 30 kilometers of walking trajectory.

## B  Image Variants

ARIA glasses are equipped with fisheye lenses, resulting in fisheye distortion in the original recordings. To facilitate the use of our dataset, we undistort these images using two different pinhole camera configurations. We provide both the original fisheye images and the undistorted versions. The three image variants are visualized in Fig. 8.

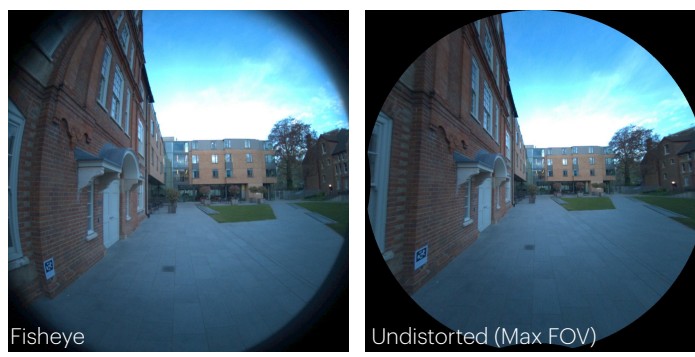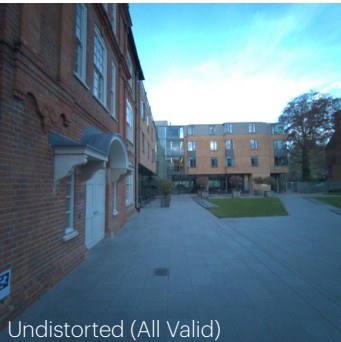

Figure 8: We provide three image types: the original fisheye, a *Max FOV* undistorted version with wider coverage and black borders (with a valid pixel mask provided), and an *All Valid* version with no black borders but a smaller field of view for easier use.

## C  Additional Details on Visual Relocalization Dataset

**Spatial Filtering.** We provide additional details about our visual relocalization dataset. In Algorithm 1, we present the pseudo-code for the spatial filtering algorithm used to eliminate redundant images when generating the database, daytime query, and nighttime query splits. For outdoor scenes (Bodleian Library, H.B. Allen Centre, Keble College, Observatory Quarter), we use thresholds of $\theta_{\text{pos}} = 1.5$ meters and $\theta_{\text{ori}} = 20°$; for the indoor Robotics Institute scene, we adopt stricter thresholds of $\theta_{\text{pos}} = 0.5$ meters and $\theta_{\text{ori}} = 20°$ to reflect its smaller scale. Notably, even after applying strong spatial filtering, our dataset includes 7,197 nighttime query images, 37 times more than the 191 nighttime queries in the Aachen Day-Night dataset [5]. Full statistics are summarized in Tab. 5.

| Algorithm 1: Spatial Filtering of Camera Poses |
| --- |

**Require:** Image list $I$ with poses $(p_i, R_i)$, thresholds $\theta_{\text{pos}}, \theta_{\text{ori}}$
**Ensure:** Filtered image list $S$
1: Shuffle $I$; initialize $S \leftarrow [\,]$, cache $C \leftarrow [\,]$
2: **for** each image $i$ in $I$ **do**
3:    $N \leftarrow \{(p_j, R_j) \in C \mid \|p_i - p_j\| < \theta_{\text{pos}}\}$
4:    **if** $N = \emptyset$ or $\forall (p_j, R_j) \in N, \angle(R_i, R_j) > \theta_{\text{ori}}$ **then**
5:      Append $i$ to $S$, append $(p_i, R_i)$ to $C$
6:    **end if**
7: **end for**
8: **return** $S$

**Coverage of Nighttime Queries Across Distance and Rotation Thresholds.** To further illustrate the challenge posed by our benchmark, Fig. 9 plots the percentage of nighttime queries that have at least one daytime database image within a specified spatial and angular threshold. Our dataset spans a wide spectrum of difficulty levels, including a particularly challenging subset: nighttime queries that are more than 5 meters and 50° away from any corresponding database image. These difficult cases account for approximately 10% of the nighttime queries. This diversity enables a more fine-grained evaluation of relocalization methods, allowing the community to assess performance across both easy and hard cases.

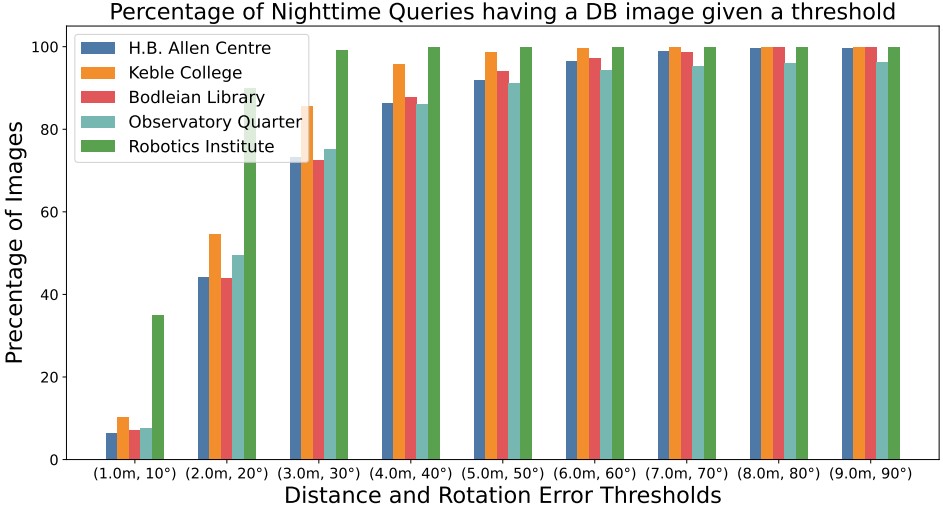

Figure 9: The percentage of nighttime queries that have a database image given a spatial and orientation threshold.

**Database Creation, COLMAP, and HLoc.** We structure our relocalization dataset using simple image lists, where each split (database, daytime queries, and nighttime queries) corresponds to a text file containing the image filenames relative to the image directory. To facilitate seamless integration with the HLoc Toolbox [48], we also provide a COLMAP model for the database images, generated using ARIA MPS output poses. Specifically, for each database image, we project the 3D point cloud of the scene onto the image plane using the corresponding ground-truth camera pose. We then apply a series of filtering steps to remove invalid projections: depth filtering, image boundary checks, and z-buffer visibility checks. From the valid set of projections, we randomly sample 3,000 2D-3D correspondences per image. Using this information, we construct the `images.bin`, `cameras.bin`, and `points3D.bin` files following COLMAP standard format. Note that our COLMAP model does not incorporate explicit occlusion reasoning. As a result, we do not recommend using it directly for PnP-RANSAC without additional filtering or refinement. However, this limitation does not affect integration with the HLoc Toolbox, as it does not rely on the database point cloud. We provide the visualization of the distribution of database, daytime, and nighttime camera poses in each scene in Fig. 10.

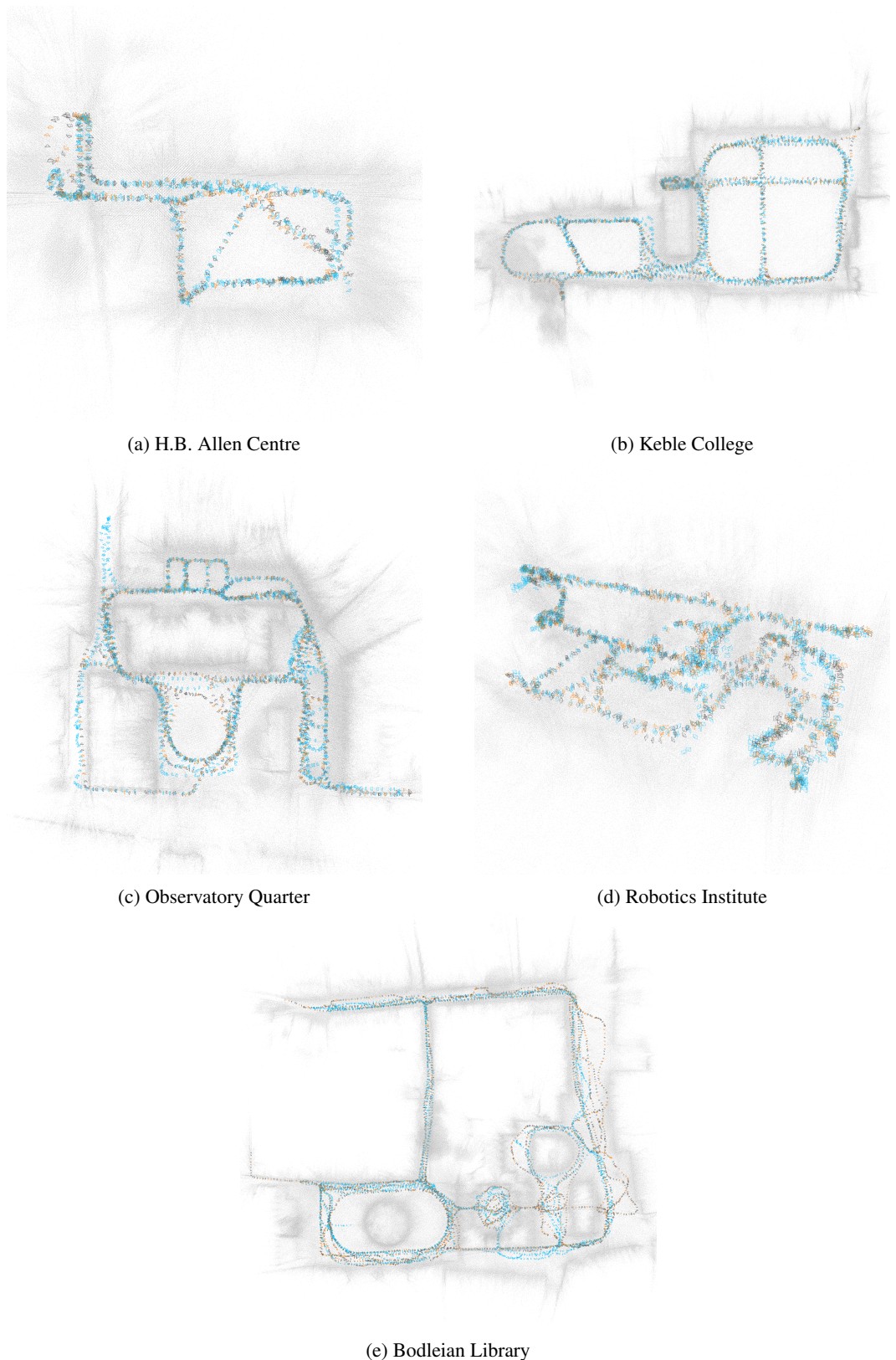

(a) H.B. Allen Centre

(b) Keble College

(c) Observatory Quarter

(d) Robotics Institute

(e) Bodleian Library

Figure 10: **Camera poses for visual relocalization in each scene.** The cameras of database images are in **black**; the cameras of day query images are in **orange** and the cameras of night query images are in **blue**.

# D Additional Results on NVS Dataset

**Image Quality.** We provide additional NVS evaluation results in Tab. 6 and Fig. 11, which complement the findings presented in Tab. 4 and Fig. 5. Specifically, Table 6 highlights that both 3DGS in-the-wild methods exhibit limited NVS performance on our dataset, as indicated by high LPIPS values. Note that PSNR and SSIM values do not capture this performance degradation.

Table 6: **3DGS In-the-Wild Image Quality.** We report image rendering quality in PSNR (↑) / LPIPS (↓) / SSIM (↑). This table complements Tab. 4 and Fig. 5 by providing additional SSIM scores.

| Method | Bodleian Library | H.B. Allen Centre | Keble College | Observatory Quarter | Robotics Institute |
|---|---|---|---|---|---|
| Splatfacto-W [46] | 25.98 / 0.60 / 0.79 | 25.65 / 0.59 / 0.81 | 27.96 / 0.59 / 0.78 | 25.83 / 0.63 / 0.78 | 22.73 / 0.61 / 0.81 |
| Gaussian-Wild [47] | 28.38 / 0.56 / 0.86 | 24.94 / 0.59 / 0.86 | 30.92 / 0.56 / 0.84 | 28.57 / 0.60 / 0.86 | 25.05 / 0.57 / 0.88 |

**Geometry.** Figure 11 visualizes the centers of Gaussian primitives after Splatfacto-W training. During this process, the initialized point cloud is culled to a reasonable density in the *H.B. Allen Centre* and *Observatory Quarter*. In contrast, the same culling procedure results in degenerate representations in the *Bodleian Library* and *Keble College* scenes, possibly due to the larger spatial extent of the *Bodleian Library* and the more extreme lighting variations present in *Keble College*.

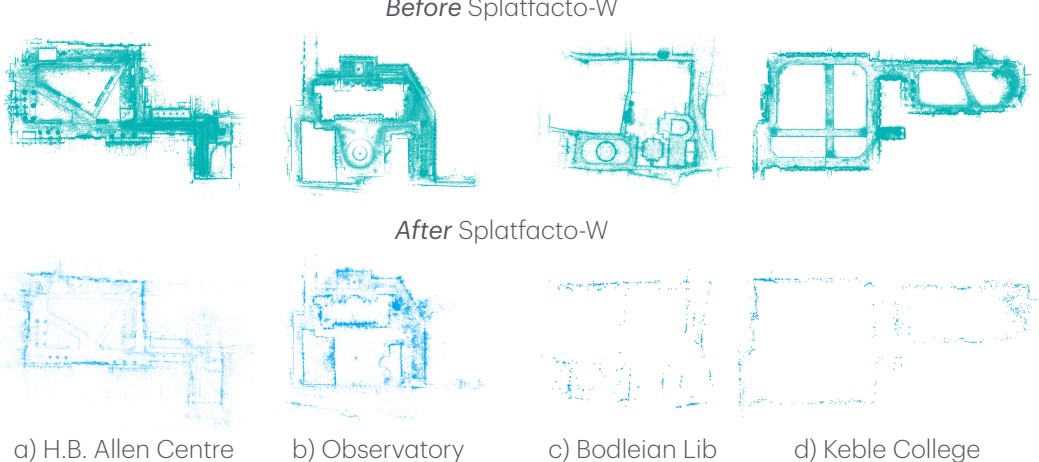

Figure 11: **Visualization of 3D Geometry.** In c) and d), less than 2000 Gaussian primitives remain after the culling process during training. This may be due to limited capability in handling large-scale scenes and dramatic light variations, resulting in a degenerated case for 3DGS rendering.

Overall, our experiments demonstrate that current 3DGS in-the-wild methods continue to face significant challenges in large-scale scenes with dramatic lighting variations.

