# OpenReview forum: "Seeing in the Dark: Benchmarking Egocentric 3D Vision with the Oxford Day-and-Night Dataset"
_NeurIPS.cc/2025/Datasets_and_Benchmarks_Track — NeurIPS 2025 Datasets and Benchmarks Track poster_

### Official Review · Reviewer_kA3F · 2025-06-11

**Rating:** 5
**Confidence:** 4

**Summary:**

This paper presents a new egocentric vision dataset for indoor and outdoor NVS and relocalization, covering a large area of scenes and long-distance trajectories. It provides day and night recordings. The provided two benchmarks of NVS and visual localization reveal the problem in the current baselines of handling egocentric and light varying capturing, which could benefit the advancement of the fields.

**Dataset Code Accessibility:**

Yes

**Dataset Code Comments:**

The dataset is already available at Huggingface with detailed instructions.

**Ethical Considerations:**

No, there are no or only very minor ethics concerns

**Final Justification:**

The rebuttal addressed my questions and I lean to aceept.

**Limitations Weaknesses:**

- Limited by the capturing device, the image quality does not seem very high compared to traditional NVS datasets like LLFF or MipNeRF360. It could be a potential concern that influences the evaluation of NVS for further usage.
- Also, the images of night seem to be more blurry and contain limited details. I'm worrying if it could heavily constrain the upper bound of the performance on the dataset due to the incomplete information. It could be useful to consider RAW images like in some HDR datasets in the future.

**Strengths Contributions:**

- The focus on large-scale egocentric data collection is significant and essential for practical reconstruction and robotics applications.
- It simultaneously supports the task of NVS and visual relocalization, which could benefit the in-depth combination of the fields to develop new methods.
- The data diversity is good, including different scenarios and light conditions. The capturing trajectories and the devices are also consistent with practical applications.

---

> ### Author Rebuttal · Authors · 2025-07-31
>
> We sincerely appreciate the positive assessment and the constructive suggestions offered.
>
> ## Image quality
> We agree that the image quality from portable glasses is generally lower than that from mobile phones or DSLR cameras due to constraints in sensor size, lens quality, and power consumption. As a result, some low-quality images may be less effective for evaluating NVS methods. While we could not fully address this issue in the current work, it reflects an inherent trade-off between dataset scale, camera pose accuracy, and image quality. Meta ARIA glasses, being lightweight and wearable, enable scalable data collection and provide robust multi-session SLAM performance through the integration of two grayscale cameras and two IMUs. While higher-quality images could have been captured separately using DSLR cameras, multi-session alignment between DSLR and ARIA devices would introduce significant complexity and potential errors. The trade-off here lies in sacrificing some image quality for scalability.
>
> ## Night images
> We agree that some night images are blurry and have limited details, due to the hardware constraints and auto-exposure configurations. Since in real-life smart glasses share similar properties, we think it might be more important to improve from image and video understanding point of view, instead of collecting a dataset with much better night image quality, which might lead to a large data distribution gap between the dataset and real-life scenario.

---

> > ### Comment · Reviewer_kA3F · 2025-08-01
> >
> > Thanks for the responses. I have no further concerns and would like to keep my positive rating.

---

### Official Review · Reviewer_UZgM · 2025-06-30

**Rating:** 5
**Confidence:** 4

**Summary:**

This manuscript presents notable contributions in egocentric, extreme illumination, and large-scale multi-session data collection, with a relatively thorough experimental analysis. However, the paper would benefit from clearer articulation of its unique advantages over existing datasets such as LaMAR, including a more systematic comparison across key attributes. The discussion of annotation errors, alignment accuracy, and potential failure cases is somewhat limited and could be expanded. Additionally, the evaluation protocols, data partitioning, and metric definitions for the benchmark tasks should be further clarified, especially regarding parameter choices and their impact on benchmark consistency. Finally, the conclusion or discussion section could more fully address the dataset’s potential for future extensions.

**Dataset Code Accessibility:**

Yes

**Ethical Considerations:**

No, there are no or only very minor ethics concerns

**Final Justification:**

I will maintain my positive opinion of this work.
It offers valuable contributions in egocentric, extreme illumination, and large-scale multi-session data collection, supported by a solid experimental analysis.

**Limitations Weaknesses:**

1.	While the differences between this dataset and existing datasets such as LaMAR are discussed, the unique contributions could be more strongly emphasized. For example, a more systematic comparison would clearly highlight the dataset’s advantages and unique features in terms of scale, acquisition conditions, diversity of illumination, and annotation quality.
2.	Although the use of Meta ARIA and SLAM techniques for multi-session data acquisition and alignment is described, the discussion on annotation errors, alignment accuracy, and potential failure cases lacks sufficient depth.
3.	For the two benchmark tasks, NVS and relocalization, the evaluation protocols, data partitioning strategies, and metric definitions should be clarified. For instance, the rationale behind specific subsampling and point cloud filtering parameters in the NVS task, as well as their impact on the consistency of the benchmark, should be explained in detail.
4.	The conclusion or discussion section could further elaborate on the dataset’s potential for future extensions.

**Strengths Contributions:**

1. The paper demonstrates a degree of innovation in egocentric, extreme illumination, and large-scale multi-session data collection.
2. The experimental analysis is relatively thorough.

---

> ### Author Rebuttal · Authors · 2025-07-31
>
> We are grateful for the reviewer’s recognition and thoughtful comments on our work.
>
> **More systematic comparison?**
>
> We have now added a systematic table (at the end of this rebuttal) comparing our dataset to existing benchmarks across several dimensions.
>
> **More discussions on annotation errors and alignment accuracy?**
>
> We will add these details in the camera ready version.
>
> **Potential failure cases?**
>
> One prominent failure case during dataset creation involves misaligned multi-session VIO trajectories. This typically occurs when there is limited trajectory overlap and extremely low lighting. We addressed this issue by performing careful manual verification and removing problematic recordings.
>
> **The evaluation protocols, data partitioning strategies, metric definitions should be clarified, For instance, the rationale behind specific subsampling and point cloud filtering parameters in the NVS task, as well as their impact on the consistency of the benchmark.**
>
> We apply subsampling primarily to reduce storage requirements and manage CPU and GPU memory usage. In our HuggingFace data release, we also provide access to the raw video recordings and point clouds. Specifically, the NVS task involves two stages of subsampling:
> 1. Video recordings originally captured at 20 fps are downsampled to 4 fps, and frames are extracted as JPEG images. This significantly reduces disk storage demands.
> 2. For each scene, the image set is further reduced to approximately 2500 frames, and point clouds are voxel-downsampled to ensure practical memory usage during training and evaluation, targeting around 100 GB for CPU memory and 80 GB for GPU memory.
>
> All benchmark experiments are conducted using this configuration to ensure consistency across evaluations. We will include further details in the camera-ready version.
>
> **Potential future extensions?**
>
> 1. Integration with semantics: While the current version of our dataset focuses on geometric tasks such as NVS and relocalisation, incorporating semantic labels (e.g., landmarks, shop names, Google Maps metadata) would enable research in semantic understanding and multimodal learning.
> 2. Scaling up: With the accessibility of Meta ARIA glasses, it is feasible to expand the dataset to more locations and include more diverse environmental conditions (e.g., varied weather and seasons).
>
>
> ---
> ### Table: Comparison of our dataset with other relevant datasets.
>
> Notes for the Table:
> - Dataset Scale: Most entries in this column are sourced from LaMAR paper.
> - TLS: Terrestrial Laser Scanning.
>
> | Dataset               | Image Collection Type | Camera Mount      | Dataset Scale                                       | Egocentric | Outdoor | Indoor | Extreme Lighting | GT Point Cloud Source |
> |-----------------------|------------------------|--------------------|-----------------------------------------------------|------------|---------|--------|-------------------|------------------------|
> | Aachen Day and Night | Sparse                 | Handheld           | ★★☆                                                | No         | ✅      | ❌     | High              | SfM                    |
> | Phototourism         | Sparse                 | Handheld           | ★☆☆                                                | No         | ✅      | ❌     | Medium            | SfM                    |
> | InLoc                | Sparse                 | Handheld           | ★★☆                                                | No         | ❌      | ✅     | Low               | TLS                    |
> | Oxford RoboCar       | Driving                | Car                | ★★★                                                | No         | ✅      | ❌     | High              | LiDAR                  |
> | CMU Seasons          | Driving                | Car                | ★★★                                                | No         | ✅      | ❌     | Medium            | SfM                    |
> | Tanks and Temples    | Walking                | Handheld           | ★★☆                                                | No         | ✅      | ✅     | Low               | TLS                    |
> | Cambridge Landmarks  | Walking                | Handheld           | ★☆☆                                                | No         | ✅      | ❌     | Low               | SfM                    |
> | ETH3D                | Walking                | Handheld           | ★☆☆                                                | No         | ✅      | ✅     | High              | TLS                    |
> | LaMAR                | Walking                | Handheld+Head      | ★★★ 3 locations, 45000 m2, 40 km, 100 hours        | Gray only  | ✅      | ✅     | High              | LiDAR                  |
> | Ours                 | Walking                | Head               | ★★★ 5 locations, 40000 m2, 30 km, 6 hours          | RGB+Gray   | ✅      | ✅     | High              | TLS                    |

---

> > ### Comment · Reviewer_UZgM · 2025-08-08
> > **response**
> >
> > I appreciate the clarifications provided and am satisfied with the revisions, so I will uphold my positive assessment.

---

### Official Review · Reviewer_fCLJ · 2025-07-02

**Rating:** 5
**Confidence:** 4

**Summary:**

The paper introduces Oxford Day-and-Night, a large-scale egocentric dataset for the tasks in novel view synthesis (NVS) and visual relocalisation. This dataset is captured using Meta ARIA glasses and processed with multi-session SLAM, spanning over a long range of recording trajectories (30km) and covering an area of 40,000 m^2. It supports benchmarking for both NVS and relocalization tasks in realistic outdoor environments.

**Dataset Code Accessibility:**

Yes

**Dataset Code Comments:**

The details of the dataset are well-documented with example data visualized for easy access.

**Ethical Considerations:**

No, there are no or only very minor ethics concerns

**Final Justification:**

The rebuttal on the key takeaways and the dataset comparison is satisfactory.

Therefore, I'm willing to raise my score.

**Limitations Weaknesses:**

Although the authors mention the limitations of existing datasets (L22-29) and differences (e.g., L51), it's a bit unclear how exactly different the proposed dataset is compared to other existing ones overall. What tasks does the existing data support? How limited the lighting diversity is (L27)? How large is the proposed dataset compared with the existing data in each task, mentioned in Related Work? I would suggest including a table summarizing missing points in the existing datasets as well as the key features of the proposed data to highlight its novelty.

Also, the authors mentioned that the experiments on the proposed dataset reveal the limitations of existing methods and underscore the value of its benchmark. However, some limitations are already exposed even with the existing datasets (e.g., as also mentioned by the authors in L253-255). What are the key takeaways that the existing datasets could not reveal in each task, but the proposed dataset did? Without such discussion, the proposed dataset can be seen as just "another" dataset, diminishing the value of the work.

**Strengths Contributions:**

The proposed data, i.e., Oxford Day-and-Night, is large with various scenes and lighting conditions. This is useful as a new benchmark dataset for future research.

There are several engineering works involved in the dataset creation with high-quality ground truths for the intended tasks. Although it's hard to see them as a technical novelty, such engineering efforts should be acknowledged by the community.

The extensive experiments on the proposed data are conducted with several existing SOTA methods.

---

> ### Author Rebuttal · Authors · 2025-07-31
>
> Thank you for acknowledging our contributions and providing valuable feedback.
>
> ---
> ## Differences Compared to Existing Datasets
> To clarify the distinctions between our dataset and existing ones, we have added a comparison table that summarizes key features and limitations of prior datasets at the end of this rebuttal. This table will be included in the camera-ready version of the paper.
>
> ---
> ## Key Takeaways
> Our dataset is designed to address major challenges in 3D perception tasks under the following conditions:
> 1. Large-scale environments
> 2. Egocentric recordings with highly agile 6-DoF motion
> 3. Drastic lighting variations across day and night
>
> Although these conditions may appear contrived, they reflect real-world challenges faced by wearable devices like smart glasses. The Oxford Day-and-Night dataset is essential for evaluating current and future algorithms under these difficult but realistic settings.
>
> Specifically, we evaluate two 3D perception tasks:
> 1. Visual relocalisation;
> 2. Novel view synthesis in the wild.
>
> Our experiments reveal substantial gaps in current methods when applied to our dataset.
>
> **Novel View Synthesis (NVS)**
>
> Most existing methods significantly struggle where there is a combination of scale, egocentric motion, and lighting variation. We observe that performance can improve when:
> - The spatial scale is reduced
> - Extremely dark images are filtered out
>
> However, even with these adjustments, the evaluated methods remain far from reliable under full dataset conditions.
>
> **Visual Relocalisation**
>
> - Extreme lighting changes severely affect most visual relocalisation pipelines.
> - We identify image retrieval as the primary bottleneck.
> - Among feature matching methods, ROMA consistently gives the best performance. Although the results achieved by ROMA are promising, there is still ample room for improvement, especially at feature matching speed.
>
> We will add these discussions to the camera ready version.
>
> ---
> ## Table: Comparison of our dataset with other relevant datasets.
>
> Notes for the Table:
> - Dataset Scale: Most entries in this column are sourced from LaMAR paper.
> - TLS: Terrestrial Laser Scanning.
>
> | Dataset               | Image Collection Type | Camera Mount      | Dataset Scale                                       | Egocentric | Outdoor | Indoor | Extreme Lighting | GT Point Cloud Source |
> |-----------------------|------------------------|--------------------|-----------------------------------------------------|------------|---------|--------|-------------------|------------------------|
> | Aachen Day and Night | Sparse                 | Handheld           | ★★☆                                                | No         | ✅      | ❌     | High              | SfM                    |
> | Phototourism         | Sparse                 | Handheld           | ★☆☆                                                | No         | ✅      | ❌     | Medium            | SfM                    |
> | InLoc                | Sparse                 | Handheld           | ★★☆                                                | No         | ❌      | ✅     | Low               | TLS                    |
> | Oxford RoboCar       | Driving                | Car                | ★★★                                                | No         | ✅      | ❌     | High              | LiDAR                  |
> | CMU Seasons          | Driving                | Car                | ★★★                                                | No         | ✅      | ❌     | Medium            | SfM                    |
> | Tanks and Temples    | Walking                | Handheld           | ★★☆                                                | No         | ✅      | ✅     | Low               | TLS                    |
> | Cambridge Landmarks  | Walking                | Handheld           | ★☆☆                                                | No         | ✅      | ❌     | Low               | SfM                    |
> | ETH3D                | Walking                | Handheld           | ★☆☆                                                | No         | ✅      | ✅     | High              | TLS                    |
> | LaMAR                | Walking                | Handheld+Head      | ★★★ 3 locations, 45000 m2, 40 km, 100 hours        | Gray only  | ✅      | ✅     | High              | LiDAR                  |
> | Ours                 | Walking                | Head               | ★★★ 5 locations, 40000 m2, 30 km, 6 hours          | RGB+Gray   | ✅      | ✅     | High              | TLS                    |

---

> > ### Comment · Reviewer_fCLJ · 2025-08-04
> >
> > Thank you for the reply.
> >
> > The provided comments on the key takeaways and the new table for the dataset comparison are satisfactory.
> >
> > Therefore, I'm willing to raise my score.

---

### Official Review · Reviewer_n6Pj · 2025-07-04

**Rating:** 5
**Confidence:** 4

**Summary:**

This works contributes a new dataset called "Oxford Day-and-Night, which is designed to benchmark NVS and Localization during light and dark environments from egocentric devices.
The dataset is captured with meta aria device and Frontier LIDAR, which are used to capture video/IMU and point cloud respectively.
 The authors align LIDAR and RGB devices using  AprilTags images and use ICP for further improvement.
They provides NVS and relocalization on their captured datasets.

**Dataset Code Accessibility:**

Yes

**Dataset Code Comments:**

I think their huggingface page provides well documents for this dataset.

**Ethical Considerations:**

No, there are no or only very minor ethics concerns

**Final Justification:**

The authors have addressed my minor concerns, and I continue to recommend acceptance.

**Limitations Weaknesses:**

I can't draw any valid conclusions from the benchmark the author set up, for example, are the significant performance drops only from the lighting? Are there other potential factors? Is there some more in-depth verification analysis?

**Strengths Contributions:**

I find this dataset to be a highly valuable contribution for the community. It offers two key advantages:
1. Scarcity of large-scale scenes with ground-truth point clouds. Datasets of this scale that include accurate, registered point-cloud ground truth are exceedingly rare, making this resource uniquely beneficial.
2. Diverse and extreme lighting conditions The inclusion of varied illumination—particularly the challenging, extreme low- and high-light scenarios—is especially appreciated, as it greatly broadens the applicability of evaluation.

Furthermore, the authors provide benchmark results for several state-of-the-art novel view synthesis and visual relocalization methods on this new dataset.

---

> ### Author Rebuttal · Authors · 2025-07-31
>
> We sincerely appreciate the reviewer’s recognition of our work and the thoughtful feedback.
>
> Regarding the conclusions and the question of whether lighting alone accounts for the observed performance drop-offs, we provide the following clarifications and insights:
>
> **Novel View Synthesis (NVS)**
>
> - We observe that reducing scene scale improves NVS performance. For instance, in the Bodleian scene, restricting training to a single street block leads to notable improvements. This suggests that handling large-scale environments remains a challenge for current NVS-in-the-wild methods and presents a promising direction for future research.
> - Additionally, excluding extremely dark images also improves results. However, in night-time scenarios, dark images are unavoidable. This highlights another potential direction: developing techniques to effectively utilize dark images, rather than simply discarding them.
>
> **Visual Relocalization**
>
> Current visual relocalization pipelines continue to struggle under extreme lighting variations.
> - Image retrieval emerges as the primary bottleneck. Among the evaluated methods, MegaLoc shows the most promise, though there is still substantial room for improvement in retrieval robustness.
> - ROMA outperforms other feature matching methods, especially under extreme lighting conditions. Despite its strong performance, it suffers from slow inference speed. Enhancing the efficiency of such feature matching approaches without sacrificing accuracy is another valuable research direction.
>
> **Other Contributing Factors**
>
> While our current analysis emphasizes lighting, additional factors such as motion blur, HDR effects, auto exposure, and ISO noise may also impact performance. We did not explore these in the current study, but we acknowledge their potential influence and leave further investigation to future work.

---

> > ### Comment · Reviewer_n6Pj · 2025-08-05
> >
> > Thanks to the authors for their hard work. I will maintain my rating.

---

### Decision · Program_Chairs · 2025-09-18

**Decision:**

Accept (poster)

**Comment:**

This paper introduces the Oxford Day-and-Night dataset, a large-scale egocentric dataset for novel view synthesis (NVS) and visual relocalization tasks. The proposed dataset provides large-scale scenes with ground-truth point clouds and these scenes have diverse and extreme lighting conditions, which are not included in previous similar datasets.
All reviewers acknowledge the contributions of this paper and agree to accept it. AC hopes that all comments will be incorporated into the final version.

I give a spotlight to this paper, because of the interesting aspects of capturing both day and night scenes, and the dataset can support evaluation on a variety of tasks.

===== FINAL UPDATE FROM DB Track PCs ====

The final decision for this paper has been taken by the program chairs after consultation with the SACs. All Senior Area Chairs have ranked papers according to the feedback from the AC during the review process. We decided to leave the original meta-review to reflect the opinion of the AC in light of the initial discussions with reviewers and SAC.